# Horsepower of Doctors’ Cars Correlates with Cardiovascular Risk and Sedentary Lifestyle but Not with Sexual Dysfunction or Sexual Satisfaction

**DOI:** 10.3390/ijerph16111932

**Published:** 2019-05-31

**Authors:** David Niederseer, Thomas Gilhofer, Christian Schmied, Bernhard Steger, Christian Dankl, Hans Peter Colvin, Josef Rieder, Daniel Neunhäuserer, Josef Niebauer, Christian Datz

**Affiliations:** 1Department of Cardiology, University Heart Center, University Hospital Zurich, Raemistrasse 100, 8091 Zurich, Switzerland; thomas.gilhofer@gmail.com (T.G.); christian.schmied@usz.ch (C.S.); 2Institute of Sports Medicine, Prevention and Rehabilitation, Paracelsus Medical University of Salzburg, Lindhofstraße 20, 5020 Salzburg, Austria; j.niebauer@salk.at; 3Department of Internal Medicine, General Hospital Oberndorf, Teaching Hospital of the Paracelsus Medical University, Paracelsusstrasse 20, 5110 Oberndorf, Austria; c.datz@kh-oberndorf.at; 4Department of Ophtalmology, Medical University Innsbruck, Anichstrasse 35, 6020 Innsbruck, Austria; bernhard@steger.tirol; 5Precise TV Ltd., 2 Riding House Street, Henry Wood House, London W1W 7FA, UK; 6Department of Ophtalmology, Paracelsus Medical University Salzburg, Muellner Hauptstrasse 48, 5020 Salzburg, Austria; j.colvin@salk.at; 7Department of Anesthesiology, Medical University Innsbruck, Anichstrasse 35, 6020 Innsbruck, Austria; J.RIEDER@tirol-kliniken.at; 8Sport and Exercise Medicine Division, Department of Medicine, University of Padova, Via Giustiniani 2, 35128 Padova, Italy; d.neunhaeuserer@gmail.com

**Keywords:** automobile, physician, lifestyle, sexual activity, sexual dysfunction, sexual satisfaction

## Abstract

**Background:** The horsepower not only of doctors’ cars correlates with personal income and social status. However, no clear relationship has previously been described between the horsepower of doctors’ cars and cardiovascular health or sexual dysfunction and/or satisfaction. **Objective:** Cross-sectional online survey to evaluate associations between self-reported horsepower of physicians’ cars and health aspects. **Methods:** Of 1877 physicians from the two University-Hospitals in Austria that were asked to participate in the study, 363 (37.7 ± 8.0 years, 208 (57.3%) men) were included into the final analysis. **Results:** Physicians that own a car with a stronger engine were significantly older, were more often male, had more often a leading position, had a higher monthly income (all *p* < 0.001), had a higher scientific output (*p* = 0.030), and had hypercholesteremia more often (*p* = 0.009). They also tended to have a higher body mass index (*p* = 0.088), reported a higher maximum weight in previous years (*p* = 0.004) and less often reported regular healthy commuting to and from work (*p* = 0.010). No significant associations were found for self-reported physical fitness, smoking status, and arterial hypertension. In addition, sexual satisfaction and sexual dysfunction were also not related to horsepower in the whole population and the male subgroup. The findings essentially persisted after controlling for age. **Conclusion:** The horsepower of Austrian physicians’ cars correlates with senior position and increased cardiovascular risk. However, our data shows no relationship between sexual dysfunction or lack of sexual satisfaction and the horsepower of doctors’ cars.

## 1. Introduction

The horsepower (HP) of a car refers to the strength of its engine. The Scottish engineer James Watt (1736–1819) first described HP to illustrate the power of steam engines compared to the power of draft horses. Depending on the definition, a mechanical HP equals 746 watts; a metric HP equals 735.5 watts [1]. Although with the implementation of the EU Directive 80/181/EEC on 1 January 2010, the use of HP in the EU is permitted only as a supplementary unit to the Système international d’unités (SI) unit watt [2], HP is still used as the standard unit in everyday life to quantify the power of a car. More HP will usually make a car accelerate faster and reach a higher top speed. Consequently, drivers of cars with more HP go faster than drivers of cars with less HP, as was recently shown [3]. 

Besides the data relating to the negative impact of excessive car driving on the environment and road accidents [4] several previous studies have addressed the relationship between car driving and health determinants, and showed that longer car driving is associated with negative health determinants such as obesity, smoking, lack of physical activity, stress, and poor mental health [5,6,7,8,9,10,11,12]. As all the existing data refers to the relationship between the duration of car driving and health aspects, we aimed to address a possible similar relationship between the amount of HP and health aspects. To our surprise and to the best of our knowledge no study so far investigated a relationship between HP and any health outcomes such as cardiovascular risk factors, sedentary life style, or socioeconomic status. We hypothesized that HP of an owner’s car is related to the hours of driving the car at least to some degree.

In addition, in popular culture an association between strong cars as means of compensation in sexual function and satisfaction has been hypothesised, however there are only questionable sources such as the “urban dictionary”, with its long-standing reputation for inner city, street, and gang slang which values extreme, startling material [13]. Even the so-called “male enhancement” industry uses "horsepower" as a metaphoric name of their erectile enhancement products [14]. Actually, there is a lack of evidence when it comes to the relationship between cars and sexual dysfunction and to the best of our knowledge, no study has previously tested the validity of this hypothesis.

It was the aim of this study to investigate the relationship of the HP of physicians’ cars with health determinants including sexual dysfunction and sexual satisfaction.

## 2. Methods 

### 2.1. Study Design and Setting

The study design has previously been described in detail elsewhere [15]. Briefly, from March until June 2005 we conducted an anonymous cross-sectional online-survey among all staff physicians of the Medical University of Innsbruck and the Paracelsus Medical University Salzburg, Austria. This survey entitled “The Innsbruck and Salzburg Physician Lifestyle Assessment, TISPLA” was based on a WHO CINDI (World Health Organisation, Countrywide Integrated Noncommunicable Diseases Intervention) survey format [16]. The questionnaire contained 170 items and was structured in different categories: personal details, sociocultural data, and health variables. Personal data included gender, age, nationality and anthropometric details (height and weight). Subsequently, body mass index (BMI, kg/m^2^) was calculated. Sociocultural information included details about family (marital status, children, etc.), religion, and frequency of health check-ups. The survey further assessed academic position, and scientific achievements as well as average monthly income. Details were asked concerning sleep, dietary and physical exercise habits, current and past medical history, health problems related to stress, sexual dysfunction and sexual satisfaction, medication and substance abuse, social status, and HP of the physicians’ cars. 

This study complies with the Declaration of Helsinki and no formal approval by an ethics committee (“ethics waiver”) was needed after correspondence with the ethics committee of the Medical University of Innsbruck. As this study is not a clinical trial, it has not been registered. A link to the online questionnaire was emailed to each registered physician in both hospitals with permission of the hospital executive directorate. Answered questionnaires were accepted for 4 weeks and coded in the order in which they were received. Data was collected strictly anonymized and no contact between the study team and any subject took place at any time, nor would it be possible due to the entire anonymous nature of the data acquisition.

We stratified the sample into four groups according to the reported HP of their car: <80 HP; 80–120 HP; 120–160 HP; >160 HP. Based on the participants’ answers in the survey jobs were divided into leading and non-leading positions. A leading position was defined as either head of department or senior consultant with academic function. Furthermore, scientific output was measured by the number of publications (first and co-author) in peer reviewed journals during the last 2 years and their impact factor. Monthly net income was asked. Concerning daily travel the physicians were asked how they do normally reach their work place and how many minutes per day they spent walking or cycling to work or home from work. 

Self-reported fitness level was assessed by a scale from 1 to 5 (1 = very good, 5 = very bad). Physicians had to state if they were suffering from hypercholesterolemia, arterial hypertension, sexual dysfunction or not. In this regard, due to the nature of this survey, no blood tests or other measurements were included. Sexual satisfaction was evaluated using a scale from 1 to 5 (1 = not satisfied and 5 = very satisfied). Additionally, the maximum weight since their 20th birthday was asked. Based on the reported smoking habits physicians were subdivided in three groups: current smokers, former smokers, and never smokers. Work stress coping strategies were assessed in how far the physicians felt overburdened in their working routine. We also asked if the physicians are or already have been on maternity leave. Wording of the questions were based on WHO CINDI (World Health Organisation, Countrywide Integrated Noncommunicable Diseases Intervention) survey format [16]. 

### 2.2. Statistical Analysis

We used GraphPad Instat 3 (GraphPad Software Inc., San Diego, CA, USA) and IBM SPSS Statistics for Windows, Version 22.0 (IBM Corp, Armonk, NY, USA) for statistical calculations. Descriptive statistics were used to assess differences for all independent variables and covariates. All data was tested for normal distribution and accordingly, Pearson’s Chi Square Test and ANOVA analysis were used to calculate statistical differences. Thereafter data were analysed corrected for age. Correlation was calculated with Pearson’s or Spearman’s correlation as appropriate. All data are reported as means ± SD or as Medians. All *p*-values are two-sided and statistical significance was set at *p* < 0.05. 

## 3. Results

### 3.1. Participants

Of 1877 physicians asked to participate in our study, 590 (31.4%) questionnaires were submitted. We excluded all incomplete questionnaires, resulting in a total of 363 (19.3%) questionnaires that were included into this study. 

Physicians without a car, or without reporting the HP of their car were excluded from the analysis (*n* = 36, see Figure 1). The age of the 208 (57.3%) participating men and 155 (42.7%) women ranged from 25 to 65 with a mean of 37.7 ± 8.0 years; subjects’ body mass index averaged 23.4 ± 3.0 kg/m^2^. The sample consisted of nine (2.5%) department chairs, 113 (31.1%) senior consultants, 45 (12.4%) junior consultants, 172 (47.4%) residents, 13 (3.6%) general practitioners working in the hospital and 11 (3.0%) other physicians. This sample is representative for the physician staff working in Austrian academic medical centres. Of 200 physicians with completed specialty training, 31 were surgeons (16%), 28 internists (14%), 27 anaesthesiologists (14%), 17 neurologists (9%), 17 paediatricians (9%), 14 gynaecologists (7%), 11 radiologists (6%), 9 dermatologists (5%), and 46 physicians trained in further specialties (23%). 

### 3.2. Associations between HP of Physicians’ Cars and Health Determinants

In 363 questionnaires HP was reported: *n* (<80 HP) = 114; *n* (80–120 HP) = 175; *n* (120–160 HP) = 53; *n* (>160 HP) = 21. Significant associations could be found for age (*p* < 0.001), gender (*p* < 0.001), job position (*p* < 0.001), scientific output (*p* = 0.03), and monthly income (*p* < 0.001). Furthermore, the association between HP and minutes per day spent with walking or cycling to work or home from work (*p* = 0.010), hypercholesteremia (*p* = 0.009), and maximum weight (*p* = 0.004) showed statistical significance. No significant association was found for self-reported physical fitness (*p* = 0.244), arterial hypertension, (*p* = 0.118), smoking status (*p* = 0.148), BMI (*p* = 0.088), or work stress coping strategies (*p* = 0.294). 

Additionally, our survey did not show any significant association between the self-reported sexual satisfaction and the power of the physician’s car engines, neither in the whole study population (*p* = 0.455) nor in the male subgroup (*p* = 0.527); the same was true for sexual dysfunction (*p* = 0.191 and *p* = 0.203, respectively) (see Table 1). Findings remained essentially unchanged after controlling for age.

## 4. Discussion

The Innsbruck and Salzburg Physicians Lifestyle Assessment (TISPLA) is a cross-sectional study among staff physicians of the Medical University of Innsbruck and the Paracelsus Medical University of Salzburg, Austria. It was aimed to analyze the correlation between health status, cardiovascular determinants, sexual dysfunction, lack of sexual satisfaction, and the power of the physicians` car engines. Our results show that physicians’ cars are associated with some parameters of health linked to a sedentary lifestyle including hypercholesteremia but not with sexual dysfunction or sexual satisfaction. After controlling for age, the findings remained essentially the same, except for arterial hypertension that was also significantly associated with a high HP.

Several other studies investigated the relationship between car driving time and health aspects. For example, Ding et al. [5] found that longer driving time to work or other destinations was associated with higher odds for smoking, shorter sleep duration, and insufficient daily physical activity leading to obesity and worse physical and mental health. The relationship between weight gain and commuting by car was also shown by Sugiyama et al. [6,7], Kawada [8] and Frank et al. [9] proposing a reduction of the time spent in a car as an effective health intervention. Addressing the mental health issue Christian used survey data to show a significant association between longer automobile commuting duration and less sleep as well as less time spent with spouse, children, and friends [10,11]. Driving induced stress by becoming angry at other drivers on the road, being stuck in traffic congestions or road constructions as well as finding a parking place has been investigated in a survey by Rasmussen et al. [12].

Although the relationship between the time spent in a car with health parameters has been previously investigated, the association between the HP of the owner’s car and health paramaters has, to the best of our knowledge, not been yet addressed. We speculate however, that persons with a stronger car use this car more often. This speculation is supported by two aspects, first by the fact that previous research, as indicated above, has shown that increased time spent in the car is associated with comparable negative health outcomes. Secondly, we report that those doctors with cars with less HP also commuted to work in a healthy way (i.e., in omitting the car and walk or cycle to and from work).

The sociological reasons for buying a specific car was investigated previously by Lansley where he reported on data of England and Wales in 2011. It revealed that each car segment is uniquely distributed across London, and the rest of England and Wales. The patterns were then compared with key 2011 Census variables on socio-economics to understand the extent to which spatial patterns of broad car characteristics correspond with variances in indicators of social make-up. Generally, car segments spatially cluster within cities and form the basis of two main groups: family cars and prestige cars (i.e., cars with a strong engine). The latter refers to the more expensive car segments that were found to be positively associated with more affluent neighborhoods largely due to the higher costs of purchasing these vehicles. Overall, this research has confirmed that there is indeed a strong relationship between socio-economics and car characteristics at a small-area level [17]. These findings show that socio-economic status correlate with the HP of the owner’s car. This is entirely in line with our findings that a more senior position or a higher monthly income is associated with a higher HP of the physician’s car.

Concerning certain aspects of health, similar to other studies mentioned above our survey identified a significant association between stronger cars and an unhealthier lifestyle of Austrian physicians. Owners of powerful cars more often suffered from hypercholesteremia, tended to have a higher body mass index, and reported a higher maximum weight since the age of twenty compared to physicians with less powerful cars. Furthermore, physicians with stronger cars tend to move less during their way to work or home from work. We were able to show that physicians with stronger cars spend significantly less time walking or cycling on the way to work or home from work compared to physicians with less powerful cars. Concerning self-reported fitness level, arterial hypertension, or smoking habits, there were no significant associations between physicians with more or less HP. 

Contrary to unscientific speculations on an inverse relationship between HP and sexual function, exemplified in the above quoted “urban dictionary”, we report no association between sexual dysfunction or sexual satisfaction and the amount of HP of the physicians’ cars. 

When interpreting our result several limitations must be taken into account. First, the response rate was rather low. However, in a questionnaire among physicians without any direct contact and without any incentives, we think that 31% response rate is comparable to previously published response rates to online surveys among physicians (e.g., 35% [18]). Second, in a survey as ours, where very private details such as sexual satisfaction, BMI, or eating habits are asked, subjects` answers will likely tend to be optimistic. Third, due to the nature of this entirely anonymous survey, no further questions to the subjects were permissible to, for example, clarify certain answers or ask further questions in the course of the evaluation of this survey. Finally, our results reflect the situation in the country of Austria, which cannot necessarily be applied to other countries where there are lower taxes on powerful cars or where gasoline is less expensive. 

## 5. Conclusion

In summary, the HP of Austrian physicians’ cars are associated with some parameters of health linked to a sedentary lifestyle including hypercholesteremia but not with sexual dysfunction or sexual satisfaction.

## Figures and Tables

**Figure 1 ijerph-16-01932-f001:**
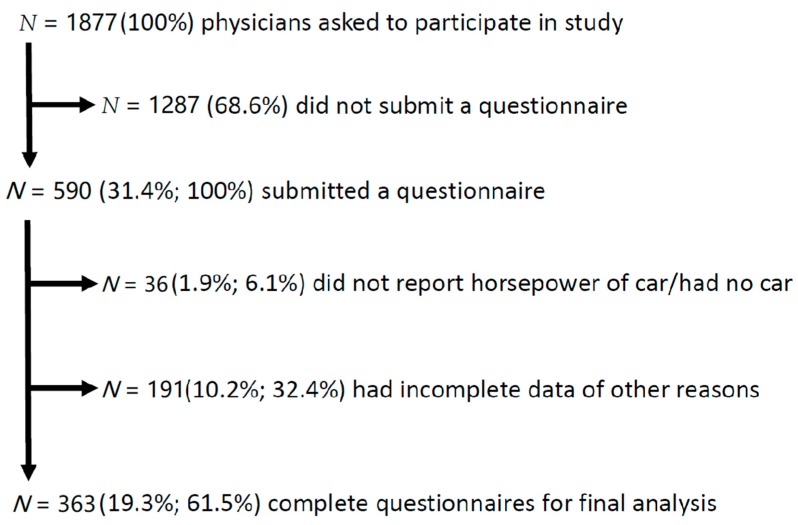
Flow chart of participating physicians of the study.

**Table 1 ijerph-16-01932-t001:** Associations of horsepower (HP) of Austrian physicians’ cars with health determinants (*n*: number; SD: standard deviation; kg: kilogram; m: meter).

Variable	<80 HP *n* = 114	80–120 HP *n* = 175	120–160 HP *n* = 53	>160 HP *n* = 21	*p* Value	*p* Value (Corrected for Age)
age (years; mean ± SD)	35.6 ± 7.7	37.8 ± 7.7	41.0 ± 8.6	39.4 ± 7.9	<0.001	n.a.
male (*n* (%))	48 (42.1)	105 (60.0)	37 (69.8)	18 (85.7)	<0.001	<0.001
leading position (*n* (%))	21 (18.4)	63 (36.0)	25 (47.2)	13 (61.9)	<0.001	<0.001
monthly income (Euro, rounded; mean ± SD)	1800 ± 300	2100 ± 500	2400 ± 400	2500 ± 400	<0.001	<0.001
scientific output (IF; mean ± SD)	26.3 ± 61.0	25.5 ± 45.5	60.8 ± 91.8	33.7 ± 42.1	0.206	0.001
scientific output (publications; mean ± SD)	8.4 ± 20.7	11.0 ± 20.6	22.4 ± 30.6	10.9 ± 17.4	0.032	<0.001
daily travel by walking/cycling (no travel, *n* (%))	30 (26.3)	48 (27.4)	24 (45.3)	8 (38.1)	0.010	0.262
fitness level (1-very good; 5-bad)	2.73 ± 0.88	2.84 ± 0.88	2.77 ± 0.98	3.24 ± 1.00	0.118	0.837
BMI (kg/m^2^; mean ± SD)	23.0 ± 3.0	23.4 ± 3.1	23.7 ± 3.0	24.7 ± 3.0	0.088	<0.001
hypercholesteremia (*n* (%))	9 (7.89%)	13 (7.43%)	8 (15.09%)	6 (28.75%)	0.009	0.001
max. weight (kg; mean ± SD)	74.4 ± 17.0	76.9 ± 16.5	79.4 ± 15.3	84.3 ± 10.7	0.004	<0.001
maternity leave (*n* (%))	23 (20.18%)	35 (20.00%)	3 (5.66%)	1 (4.76%)	0.002	<0.001
current smokers (*n* (%))	25 (22.5%)	41 (23.8%)	10 (18.9%)	9 (42.9%)	0.148	0.486
former smokers (*n* (%))	25 (22.5%)	47 (27.3%)	20 (37.7%)	3 (14.3%)	0.148	0.486
never smokers (*n* (%))	61 (55.0)	84 (48.8)	23 (43.4)	9 (42.9)	0.148	0.486
Arterial hypertension (*n* (%))	5 (4.4)	6 (3.4)	5 (9.4)	2 (9.5)	0.244	0.002
sexual dysfunction (*n* (%))	6 (12.5)	4 (3.8)	4 (10.8)	2 (11.1)	0.191	0.614
sexual dysfunction (*n* (% of male subgroup))	6 (12.8)	4 (3.9)	4 (5.6)	2 (11.1)	0.203	0.104
sexual satisfaction (1-very good, 5-bad; mean ± SD)	3.23 ± 1.36	3.36 ± 1.23	3.46 ± 1.10	3.10 ± 1.34	0.455	0.424
sexual satisfaction in male subgroup (1-very good, 5-bad; mean ± SD)	3.21 ± 1.41	3.40 ± 1.13	3.53 ± 1.16	3.11 ± 1.41	0.527	0.190

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
