# Peer review of "Horsepower of Doctors’ Cars Correlates with Cardiovascular Risk and Sedentary Lifestyle but Not with Sexual Dysfunction or Sexual Satisfaction"

_ijerph, 2019, doi:10.3390/ijerph16111932_

Round 1
Reviewer 1 Report
The authors present a well written manuscript on how the horsepower of doctors' cars are associated with various health outcomes.
INTRODUCTION
The authors do not provide an adequate justification for looking at horsepower and how it might be associated with various (unrelated) outcomes. They have an extensive section on the background of horsepower but do not justify why they are using this measure as an exposure to potentially detrimental health outcomes. Authors provide interesting information about why length of car driving (sedentary behavior) is associated with poor health, but do not tie it back into their measure. A proposed mechanism is imperative to this study that may not be intuitive in the scientific community. They also add a hypothesis about how horsepower may have implications for sexual dysfunction but is a disjointed piece from the sedentary implications of driving. Urban dictionary may not be an appropriate citation for a proposed hypothesis.
METHODS
The authors use the TISPLA study from 2005 (which may be slightly dated as the data is 15 years old).
The authors describe over 20 variables that are not related, may need more focus, and not all are justified in the introduction or discussed later in the manuscript.
How was fitness level assessed? How was the scale developed?
The authors should elaborate on their statistical analysis and potentially use methods that allow adjustment for potential confounders and narrow their focus to specific outcomes.
DISCUSSION
Again, the authors have not linked horsepower with a sedentary lifestyle. They have one sentence "We speculate however, that persons with a stronger car use this car more often" but their supporting results do not support that speculation.
The authors also speak at great lengths about the sociological reasons for buying a specific car outlined in Lansley. But this study does not support their findings or put it into context.
If the authors narrow the focus of their analysis and manuscript on related health outcomes (sexual dysfunction and cardiovascular outcomes related to sedentary behavior are not related) and provide adequate justification and a proposed mechanism for their exploration, it could be an interesting contribution to the existing literature.
Reviewer 2 Report
To say the least, this is an unusual paper, with an appeal to practitioners and other readers interested in risk factors, especially correlation with socioeconomic position (SEP). There seem to be two parts regarding strength of results, as summarized in Table 1. Here the variables of interest are presented, and the aim was to elucidate the associations of the list, mostly risk factors, with horsepower of physicians’ cars.
One thing to keep in mind is that no data was measured, but rather are collected from a questionnaire. This immediately opens the possibility of several different kinds of bias, especially memory bias, social desirability bias, and bias to avoid conflict of reality (which is not measured) with participants’ own view of his sexual satisfaction and performance (which is chiefly what is reported). However, it is more complex because the participants were physicians, who, for their own reasons and greater understanding of the implications of their health values, may have wanted to minimize certain aspects of their history. This may have operated on an unconscious level. Certainly looking at Figure 1, the extent of cooperation and performance was not impressive, with poor returns, and the attitude may have not simply been quantitative in responses or their completeness, but also qualitative.
In NHANES and other large surveys, the greatest deviation from the truth is in self-estimation of BMI and exercise, and these may actually rise to over 35% lower values reported for BMI, exercise being overestimated by up to 500%. Of all factors of interest, sexual dysfunction and sexual satisfaction have the largest personal overlay of belief, and be subject to immense error in reporting. The definitions are imprecise and the assessment was not varied or granular enough to explore these variables accurately or with objective fidelity. Basically the research tools chosen for classification were inadequate to ensure validity. How many male physicians would volunteer they are impotent, or even had lost interest in sex, when they were top earners in high positions? I think not many, particularly when the information might be disclosed, even in error, subsequently. Each man has his own definition of what “sexually satisfied” means. In contrast , the other variables are measured frequently and reported in numbers, without any emotional or judgmental implications. Most men (and physicians) have achieved their status because of some drive to do so, and one would suspect that they would be eager to divulge their accomplishments and minimize socially unacceptable qualities of obesity, sloth, and smoking. In fact, physicians who would have high powered cars would, for the very same reason, be loath to report sexual problems. In this study, where such an association was being sought, relying on self-reporting is problematic.
Erectile dysfunction (ED) results from disease of the penile microvascular bed and correlates with hypertension, hypecholesterolemia, and atherosclerosis in other vascular beds. This is an accepted, fairly universal finding. The data in this study do not appear to reflect this association, requiring explanation—as offered above. In any case, methodological shortcomings with respect to the sex topics limit the overall validity of the study.
The comments about penile size and references 13 and 14 do provide a certain levity which is novel. The Urban Dictionary (UD) has a long-standing reputation for inner city, street, and gang slang which values extreme, startling material. Their relationship with true scholarly linguists who study word usage and etymologist might be likened to the relationship of quality news reporting with tabloids. They are both sensationalistic, designed to elicit shock and clicks. For instance, “research paper” is defined as “all forms of evil brought together in the form of a paper you gather pointless information.usually assigned by the most obnoxious teacher that no one likes.”
In the culture of this dictionary, it is also common to call men "small-penised" as a baseless insult for not joining a dangerous act, a violent gang, not brandishing a gun in a nightclub, or shying away from assaulting another male who is "disrespectful". It is a general remark of rejection from a male point of view, even referring to someone who does not take the initiative, akin to “pussy” [https://www.urbandictionary.com/define.php?term=Don%E2%80%99t%20Be%20A%20Pussy ] as or feminine, weak, as used in Mr and Mrs Smith (2005 movie) at https://www.celebquote.com/2545. Specifically, the Urban Dictionary entry for “small penis” is “This guy almost always wears tighty whities or bikini underwear. Many times this guy whould have gotten a few swirlies by the end of high school”… and “new studies in 2012 find that population in anglo-saxon-european countries are on the decline because white women are becoming less receptive to white men for sexual reasons. Now the white race faces threat of extinction because of the white man's small penis.” Obviously the UD is a mixture of banter, fantasy, and for a few people, their own brand of truth.
In summary, the potential for bias, multiple confounders (including mental/behavioral), different interpretations of the sexually-related variables and their inadequate measurement, are a result of study design. Whether there is a correlation of HP with sexual prowess remains somewhat uncertain. Admittedly, assessing libido is a difficulty encountered frequently by investigators.
On the other hand, the other correlations are supported by a robust parallel literature.
A limitations section should be added, and the items discussed above should be amplified. If further data are planned, some other variables might be clarifying—the age of the wife, the number of wives, how many miles are driven, having the wife (or girlfriend, or parner) answer the sexual questions as well as the physician, This review suggests revising the section about the UD somewhat.
Other variables could include any prior consultations with urologists for ED, whether testosterone is being taken, calcium artery scores, family history of heart disease, fasting plasma insulin level, blood CRP level, penile plethysmography or PET scan of the penis to make the study rigorous. Even so, the question of unaddressed physical and mental confounders would still exist that impact each participant’s own image of his sexual vigor and health.
Round 2
Reviewer 1 Report
I have no further comments for the authors. I believe they have addressed most concerns the reviewers brought up.
Reviewer 2 Report
Sufficient improvements have been made, with most of the referee's comments addressed adequately.
The design is rate limiting. However, the authors have added selectively to correct sections where it can be done.
The authors have presented much material in their rebuttal. More of this should have been included in the discussion for optimized explanations.
All things considered, and considering the novelty and originality of the manuscript, I conclude it deserves publication.